# Exploring the Role of Miniemulsion Nanodroplet Confinement on the Crystallization of MoO_3_: Morphology Control and Insight on Crystal Formation by In Situ Time-Resolved SAXS/WAXS

**DOI:** 10.3390/nano13061046

**Published:** 2023-03-14

**Authors:** Francesca Tajoli, Maria Vittoria Massagrande, Rafael Muñoz-Espí, Silvia Gross

**Affiliations:** 1Dipartimento di Scienze Chimiche (DiSC), Università degli Studi di Padova, Via Marzolo 1, 35131 Padova, Italy; 2Institute of Materials Science (ICMUV), Universitat de València, Catedràtic José Beltrán 2, 46980 Paterna, Spain; 3Karlsruher Institut für Technologie (KIT), Institut für Technische Chemie und Polymerchemie (ITCP), Engesserstrasse 20, 76131 Karlsruhe, Germany

**Keywords:** molybdenum oxide, miniemulsion, confined space, size and shape control, nonclassical crystallization

## Abstract

Enclosed nanoscale volumes, i.e., confined spaces, represent a fascinating playground for the controlled synthesis of inorganic materials, albeit their role in determining the synthetic outcome is currently not fully understood. Herein, we address the synthesis of MoO_3_ nano- and microrods with hexagonal section in inverse miniemulsion droplets and batch conditions, evaluating the effects of spatial confinement offered by miniemulsion droplets on their crystallization. Several synthetic parameters were systematically screened and their effect on the crystal structure of h-MoO_3_, as well as on its size, size distribution and morphology, were investigated. Moreover, a direct insight on the crystallization pathway of MoO_3_ in both synthetic conditions and as a function of synthetic parameters was provided by an in situ time-resolved SAXS/WAXS study, that confirmed the role of miniemulsion confined space in altering the stepwise process of the formation of h-MoO_3_.

## 1. Introduction

In recent years, nanoscale hosts have been successfully employed as nanoreactors for the spatially and phase-controlled synthesis of inorganic systems, possibly enabling a precise control on the reaction pathway as well. Indeed, confinement into nanoscale environments has been reported to affect the physical properties of constrained molecules, as well as the mechanisms and thermodynamics of the syntheses that are performed within them [1,2,3]. Consequently, synthetic strategies exploiting these fascinating environments are particularly promising for addressing the synthesis of inorganic systems with a controlled size and shape. Within this framework, inverse (water-in-oil) miniemulsion (ME) nanodroplets represent an exciting nanostructured reaction environment. Inverse MEs are produced by application of high shear forces (e.g., ultrasounds, US) to a mixture of an excess of oily phase and an aqueous phase, in the presence of a small amount of surfactant and of an osmotically active agent (i.e., a species soluble only into the dispersed aqueous phase). ME nanodroplets (30–500 nm) are critically stabilized towards collisions by the surfactant and stabilized towards Ostwald ripening by the employed osmotically active agent (i.e., commonly the precursor’s metal salt in inorganic synthesis), that efficiently suppresses Ostwald ripening by counterbalancing Laplace pressure arising from their small size. ME nanodroplets are considered to be independent from each other, as no inter-droplet reactant exchange occurs, and they maintain their identity up to several days; each ME droplet can be seen as a nanoreactor, and reactions occur in a parallel fashion within each nanodroplet [4,5]. Several successful reports of miniemulsion synthesis of different classes of inorganic compounds, encompassing metal nanoparticles and alloys [6], metal oxides [7,8,9,10,11,12], hydroxides [13], sulfides [14], and fluorides [15], can be found in the literature, although a thorough investigation of the effects of ME space confinement on the synthetic outcome has not been reported. Within this framework, and with the aim of exploring the unique control offered by ME nanodroplets on the size and shape of the synthesis outcome, and possibly on the reaction pathway, in this work we address the miniemulsion confined synthesis of molybdenum(VI) oxide. Moreover, going beyond the state-of-the-art, we present a systematic investigation of the role of spatial constraint offered by ME droplets on the crystallization of MoO_3_ by comparing the results obtained within ME droplets with those by batch approach (i.e., macroreactor), as detailed below.

Molybdenum(VI) oxide, a wide band gap semiconductor employed in various fields of application (e.g., photocatalysis [16,17,18], photochromic and electrochromic devices [19,20,21], and gas sensors [22,23]), is known to crystallize in three main polymorphs, depending on the crystallographic arrangement of MoO_6_ building blocks: α-MoO_3_, β-MoO_3_, and h-MoO_3_. The thermodynamically stable orthorhombic α-MoO_3_ (space group *Pnma*) is characterized by a 2D layered structure of zigzag chains of MoO_6_ distorted octahedra sharing edges and corners [24]. The metastable monoclinic β-MoO_3_ (*P2_1_/n*) has a ReO_3_-type structure in which the MoO_6_ octahedra share corners to build a distorted cube [25]. On the other hand, the metastable hexagonal h-MoO_3_ (*P6_3_/m*) is constituted by zigzag chains of MoO_6_ octahedra, giving rise to a crystalline structure with large 1D tunnels (ca. 3.0 Å in diameter) with hexagonal symmetry along the *c*-axis [26,27]. Because of the zigzag structure formed by sharing edges and consequent stronger bonding of MoO_6_ octahedra along the *c*-axis, h-MoO_3_ usually shows high anisotropy, resulting in the growth of rods with hexagonal cross-sections [27]. The peculiar tunnel-structure of h-MoO_3_ may result into enhanced properties of this metastable polymorph when compared to the other ones. For instance, ion exchanges in h-MoO_3_ are facilitated and electron-hole separation under irradiation is accelerated, ensuring a better photocatalytic activity towards the degradation of organic pollutants with respect to α-MoO_3_ [17,21,28].

Molybdenum oxide, possibly hydrated, can be synthesized via chemical precipitation by the acidification of Mo(VI) solutions [16,27,29,30,31] or via the decomposition of peroxomolybdate precursors [21,32,33,34]. Its hexagonal polymorph h-MoO_3_ is often prepared in hydrothermal conditions [17,21,30,33,34], or, in general, the reported procedures involve heating the reaction mixture above 70–100 °C [16,27,29,31]. On the contrary, herein we address the synthesis of crystalline h-MoO_3_ at room temperature both in batch conditions (i.e., simple mixing of the precursors) and in the confined space of inverse miniemulsion droplets, to unveil the effects of ME spatial constraint. Our thorough investigation of the effects of space confinement on h-MoO_3_ synthetic pathway and reaction outcome (i.e., crystalline structure, size shape and aggregation) was also supplemented by a systematic screening of the main synthetic parameters with both synthetic approaches, and supported by both ex situ (XRD, ATR, TEM, and SEM) and in situ time-resolved (simultaneous SAXS and WAXS) investigations at synchrotron radiation source.

## 2. Materials and Methods

### 2.1. Chemicals

Ammonium heptamolybdate tetrahydrate (AHM, (NH_4_)_6_Mo_7_O_24_·4H_2_O) was purchased from Alfa-Aesar. Concentrated nitric acid (HNO_3_, 65 wt%), cyclohexane and Span80 were purchased from Sigma-Aldrich (Darmstadt, Germany). Polyglycerol polyricinoleate (PGPR) was purchased by Palsgaard. All chemicals were used without further purification.

### 2.2. Miniemulsion (ME) Approach

A mixture of an aqueous solution of AHM with different concentrations (0.10, 0.15, 0.20, 0.25 M) and a solution of 1.5 wt% Span80 in cyclohexane (aqueous phase:oil phase 1:3 wt) was homogenized by ultrasonication (Sartorius Stedim LabsonicP, 24 kHz, 280 W, 0.9 s pulse) until a white turbid ME was obtained. To the as-prepared ME, an excess of conc. HNO_3_ (65 wt%) was added dropwise, screening different AHM:HNO_3_ molar ratios (1:10, 1:15, 1:20, 1:25 mol). After acid addition, a further ultrasonication (280 W, 0.9 s pulse, 2 min) of the reaction mixture was carried out. A series of syntheses in which this last synthetic step was not performed was also carried out (see Appendix A for samples list). The resulting mixture was allowed to stand at room temperature for 5 min, 6 h, 18 h or 24 h, after which the ME was coagulated with acetone. The precipitate was then isolated and purified by centrifugation with acetone and deionized water (12,000 rpm, 5 min), dried under vacuum at room temperature and finely ground for analyses.

### 2.3. Batch Approach

An excess of conc. HNO_3_ (65 wt%) was dropwise added to an aqueous solution of AHM (final pH < 1), under mechanical stirring and at room temperature. Different concentrations of the precursor solution (0.10, 0.15, 0.20, 0.25 M) and different AHM:HNO_3_ molar ratios (1:10, 1:15, 1:20, 1:25 mol) were screened. A white precipitate was immediately formed, and the suspension was left stirring for 5 min, 6 h, 18 h or 24 h. The precipitate was then isolated and purified by centrifugation with deionized water (12,000 rpm, 5 min), dried under vacuum at room temperature and finely ground for analyses.

### 2.4. Experimental Setup for the In Situ Time-Resolved SAXS/WAXS Study

A schematic representation of the continuous-flow setup built at the SAXS beamline at Elettra Sincrotrone Trieste is reported in Figure 1. A reaction vessel equipped with an external jacket connected to a circulating water bath was employed to keep the reaction mixture cooled (T ≈ 20 °C). The vigorously stirred reaction mixture was funneled through a continuous-flow equipment (Viton^®^ tubing, i.d. 3 mm, wall th. 1 mm) with the aid of a peristaltic pump (Ismatec MCP) into a quartz capillary (o.d. 1.5 mm), where SAXS and WAXS profiles were acquired simultaneously every 10 s, as detailed below The total residence time in the flow system was 7 s. Inverse MEs were produced at the beamline using a remotely controlled Sartorius Stedim LabsonicP homogenizer (US: 2 min, 280 W, 0.9 s pulse). HNO_3_ was injected into the reaction mixture by using a remotely controlled syringe pump (Harvard PHD 4400, 1 mL min^−1^ flow rate). The evolution of the scattering profile of the reaction mixtures was followed starting from 40 s before ME production (for ME series) or 40 s before acid injection (for batch series). For either case, the start of acid injection was set as t = 0 s and the evolution of the scattering profile was followed up to 1–3 h, depending on the observed variations in pattern features.

### 2.5. Characterization Methods

XRD characterization was carried out with a Bruker D8 AXS Advance Plus diffractometer, equipped with a Cu-Kα_1,2_ anode (*λ* = 1.5406 Å, supplied with 40 kV and 40 mA) and mounting a LYNEXEYE XE-T detector employed in 1D mode. Data were collected with a Bragg-Brentano geometry (*θ*–2*θ*) over the 10–90° 2*θ* range (step size: 0.026° 2*θ*, nominal time: 0.5 s per step), with fixed divergence slits of 0.50° and Soller slits with an aperture of 2.5°. Data were analyzed with the Bruker Diffrac. Suite. Crystalline phases were identified with the software DIFFRAC.EVA. Crystallographic and microstructural information of MoO_3_ were estimated by fitting the experimental diffraction patterns using a WPPF method implemented in the DIFFRAC.TOPAS software [35].

TEM characterization was performed using a Tecnai G^2^ (FEI) instrument, operating at 100 kV. Images were acquired with a Veleta digital camera. Samples were prepared by depositing one droplet of powder suspension in ethanol on a 400-mesh holey film Cu grid. The dimensional analysis of both length and width of MoO_3_ rods on the obtained micrographs was manually performed using Fiji software [36,37], and the aspect ratio was calculated as the ratio between their width and length.

SEM measurements were performed using a Field Emission (FE-SEM) Zeiss SUPRA 40VP, equipped with an Oxford INCA x-sight X-ray detector, employing a primary beam acceleration voltage of 5.0 kV and a secondary electron detector. Samples were prepared by drop casting 3–5 droplets of powder suspension in ethanol on a Si wafer.

ATR experiments were performed with a Nicolet Nexus 870 equipped with an ATR accessory using a diamond crystal. ATR spectra were acquired in the 4000–500 cm^−1^ range, collecting 32 scans with spectral resolution of 4 cm^−1^.

DLS analyses on the as-prepared ME of Mo precursor were carried out using a Malvern Zetasizer NanoS, mounting a He-Ne laser (*λ* = 633 nm) operated in backscattering mode (i.e., collection angle of 173°).

Simultaneous in situ SAXS and WAXS measurements were performed at the SAXS beamline at Elettra Sincrotrone Trieste by setting X-rays energy to 8 keV (i.e., 1.54 Å), by employing a Si(111) monochromator, and focusing the beam with a double toroidal mirror on the measuring quartz capillary. SAXS profiles were collected with a 10 s time step (8 s acquisition time), by employing a 2D Pilatus3 1M detector, positioned after a 1 m long vacuum flight tube. The collected 2D patterns were azimuthally and radially integrated into 1D plots of the scattering function, $I\left(q\right)\;\mathrm{vs}.\;q$. The final reciprocal space window available ranged from 0.1 to 7.1 nm^−1^ (i.e., 0.1–10.0° 2*θ*). WAXS profiles were collected simultaneously with a 2D Pilatus 100k detector, positioned with a tilting of approximately 30° with respect to the vertical axis, to result in an accessible q-range of 13.4–23.9 nm^−1^ (i.e., 19–34° 2*θ*). The angular scale of the measured intensity was calibrated using silver behenate and *p*-bromobenzoic acid as the standards for SAXS and WAXS regimes, respectively. Data were corrected for primary intensity fluctuations and the corresponding background (i.e., either water or cyclohexane for batch and ME syntheses, respectively) was subtracted from each scattering pattern.

## 3. Results and Discussion

### 3.1. Synthesis of Hexagonal MoO_3_ by Inverse Miniemulsion and Batch Approaches

For the synthesis of molybdenum(VI) oxide by both inverse miniemulsion and batch approaches, the dehydration and polycondensation reactions occurring in acidified solutions of Mo(VI) were exploited [38]. Briefly, an excess of concentrated nitric acid was added to aqueous solutions of ammonium heptamolybdate (AHM) at room temperature, and MoO_3_ was precipitated. While the batch approach involved the simple mixing of the two precursors (dropwise addition of HNO_3_ to AHM solution), the ME approach was performed by the addition of conc. HNO_3_ to a pre-formed inverse miniemulsion of AHM and subsequent further ultrasonication to promote the acid diffusion into ME droplets (i.e., diffusion method) [5,39]. A systematic variation of different synthetic parameters was carried out, within both synthetic approaches. In particular, the following parameters were screened: (i) the AHM molar concentration (0.10, 0.15, 0.20, and 0.25 M), (ii) the AHM:HNO_3_ molar ratio (1:10, 1:15, 1:20, and 1:25), (iii) the application of ultrasounds after the acid addition (see below), and (iv) the reaction time (5 min, 6 h, 18 h, and 24 h). As “reference” miniemulsion and batch samples, the following synthetic conditions were employed: [AHM] = 0.2 M, AHM:HNO_3_ of 1:10 mol, reaction time of 24 h.

XRD analysis of both the “reference” batch and miniemulsion samples showed that molybdenum (VI) oxide crystallized as the metastable hexagonal polymorph, h-MoO_3_, without the formation of any secondary products (Figure 2a,f). Interestingly, it was found that NH_4_^+^ and H^+^ cations were accommodated in h-MoO_3_ hexagonal framework, since the experimental diffraction patterns better matched the crystalline structure of (NH_4_)_0.944_H_3.304_Mo_5.292_O_18_ (ICSD 01-083-1175) rather than pure MoO_3_ (JCPDS 00-021-0569), in agreement with literature reports. Indeed, the presence of monovalent cations (e.g., Na^+^, K^+^, H^+^, and NH_4_^+^) in the cavity of the hexagonal structure of metastable h-MoO_3_ stabilizes the tunnel oriented along the *c*-axis, and they may act as structure directing agents for the formation of elongated structures [17,21,29]. Moreover, the mismatch of the relative intensities of the (h00) experimental reflections with the ones reported in the databases (insets in Figure 2a,f) indicated that the samples were characterized by a clear preferred orientation along the [100] direction, likely along the tunnel structure. The enhancement of the (h00) family of reflections’ intensity with respect to the reference ones was found more evident for ME samples than batch ones. The fitting of XRD patterns (Figure 2a,f) estimated average crystallite sizes in the 150–200 nm range and in the micron-range for ME and batch samples, respectively, confirming a certain control on the material growth in ME conditions (effect of space confinement). These observations were found consistent with TEM and SEM analyses (see some representative micrographs, rod length and aspect ratio distributions in Figure 2b–e,g–j), showing that both ME and batch products crystallized as rods with hexagonal section, as expected for the h-MoO_3_ polymorph. Notably, the size of the rods and their aspect ratio (AR, i.e., the ratio between the width and the length of the rods) were strongly affected by the synthetic strategy: on average smaller (ME: average rod length of 650 nm and width of 170 nm vs. batch: average rod length of 1.7 µm and width of 730 nm) and more elongated (ME: AR = 0.3 vs. batch: AR = 0.4) rods were obtained by confining the synthesis of MoO_3_ in ME nanodroplets. It is noteworthy to outline that, to the best of our knowledge, no smaller MoO_3_ rods than the herein discussed ME ones were reported in literature, as micron-sized rod lengths were generally obtained [16,21,27,30,33,40,41,42,43]. The agreement between TEM and XRD results, both in terms of relative average size (larger for batch samples) and aspect ratio (smaller for ME samples, confirming the more pronounced preferred orientation observed by XRD), was not found in terms of absolute particle size: for both samples, TEM average size of the h-MoO_3_ particles was larger than XRD crystallites sizes, indicating that the rods were polycrystalline. However, since TEM and SEM investigations showed well-faceted rod-like particles with a cross section resembling the crystalline symmetry of h-MoO_3_, it could be argued that the aggregation of the primary particles occurred in an oriented fashion, and the resulting rods could likely be mesocrystals (i.e., obtained by the oriented aggregation of primary particles in a common crystallographic register) [44,45].

Finally, ATR measurements (Appendix A in the SI) confirmed the effectiveness of the purification method in removing surfactant moieties in ME samples: no vibrational bands ascribable to Span80 were found in the 4500–500 cm^−1^ range of IR spectra. Moreover, along with characteristic Mo-O stretching vibrations for h-MoO_3_ (ν_O-Mo-O_ 500–600 cm^−1^, ν_Mo = O_ at 892 and 975 cm^−1^) [43], stretching and bending vibrations of N-H and O-H were observed (ν_N-H_ at 3204 cm^−1^, δ_N-H_ at 1426 cm^−1^, ν_O-H_ at 3430 cm^−1^, and δ_O-H_ at 1616 cm^−1^) [46], confirming the presence of ammonium ions and water in the structure.

#### 3.1.1. Effect of AHM Concentration

According to the LaMer model of particles formation and its quantitative implementations [47,48,49], a variation of the precursor concentration enables to experimentally tune the number of formed primary particles, and consequently their final size. Moreover, the amount of osmotic agent (i.e., herein AHM) was reported to influence the size of the ME droplets, ultimately tuning the constrained conditions the formed particles are subjected to. For these reasons, a systematic variation of the concentration of the AHM aqueous solution from 0.10 M to 0.25 M was carried out in both ME and batch conditions, keeping constant other reaction parameters (AHM:HNO_3_ of 1:10 mol and reaction time of 24 h).

The formation of metastable h-MoO_3_ with preferred orientation along the [100] direction by both ME and batch approaches, and with all synthetic conditions, was observed by XRD (Figure 2a,c). A slight increase in the crystallinity of ME samples with increasing AHM concentration was noticed, while no significant changes in the diffraction patterns of batch samples were observed (Figure 3a,c). On the other hand, the effects of the precursor concentration on the size and morphology of MoO_3_ particles were slightly more evident (Figure 3b,d). Generally, smaller and more elongated rods by ME with respect to the batch approach were observed in all the screened conditions. No clear trend was found in the average length of batch MoO_3_ rods as a function of AHM concentration, while an increase in ME rod mean length with increasing AHM concentration could be appreciated (up to 80%), in agreement with an increase in ME droplets size by increasing the concentration of the osmotic agent. A clearer trend as a function of AHM concentration was observed in the aspect ratio of both ME and batch MoO_3_ rods: by increasing the precursors concentration (with constant AHM:HNO_3_ molar ratio), the rods AR decreased (i.e., more elongated). This effect was particularly marked for batch samples, obtaining quasi-cubic micro-particles at low AHM concentration, and more elongated (AR = 0.5) prisms at higher AHM concentration (Appendix A). This observation could be rationalized considering the favored growth in length of MoO_3_ rods with respect to their growth in width due to the stronger bonding of MoO_6_ octahedra along the *c*-axis [27] and the likely faster precipitation of MoO_3_ with higher precursor concentration [47,48,49].

#### 3.1.2. Effect of AHM:HNO_3_ Molar Ratio

Since a minor amount of amorphous material in addition to the metastable h-MoO_3_ polymorph was observed in the ME samples diffraction pattern, the added acid amount was increased, screening the AHM:HNO_3_ molar ratio between 1:10 and 1:25 within both approaches and keeping the AHM concentration constant to 0.20 M, to improve the crystallinity of the product. The disappearance of the amorphous halo from 20° to 40° 2θ (up to an 80% decrease) in the ME samples XRD pattern (Appendix A) was indeed observed by increasing the acid amount, indicating an increased crystallinity of h-MoO_3_, while no appreciable differences were found in the batch samples diffractograms, being batch h-MoO_3_ highly crystalline already with AHM:HNO_3_ of 1:10 mol. In addition to affecting the crystallinity of h-MoO_3_, the AHM:HNO_3_ molar ratio was also found to play a role in determining the size and morphology of MoO_3_ nano- and micro-structures (for ME and batch samples, respectively). In particular, with higher amounts of nitric acid, ME MoO_3_ rods were characterized by sharper facets and fewer independent non-aggregated rods were observed by electronic microscopies as compared to the reference sample: they were likely to grow stacked one on top of the other, possibly as an effect of superficial charge (Figure 4a). The variation of the AHM:HNO_3_ molar ratio had a more significant effect on batch MoO_3_ morphology: evidently smaller and more elongated rods were obtained by increasing the acid amount (Figure 4b and Appendix A), possibly because of the combined effect of the likely faster reaching of supersaturation with higher amounts of precipitating agent and the well-known favored growth in length of the hexagonal MoO_3_ rods.

#### 3.1.3. Effect of Ultrasounds on HNO_3_ Diffusion into Miniemulsion Droplets

The employed diffusion approach of the ME method (i.e., external addition of the precipitating agent to a pre-formed ME containing the metal precursor) usually involves a step of homogenization of the reaction mixture (i.e., pre-formed ME and added aqueous solution of precipitating agent) to trigger the diffusion of the precipitating agent into the ME droplets. With the aim of investigating the spontaneous diffusion of nitric acid into the pre-formed ME nanodroplets containing Mo precursor, ME synthesis without applying ultrasounds (US) to the reaction mixture after the acid addition was carried out ([AHM] = 0.20 M, AHM:HNO_3_ 1:10 and 1:20 mol).

XRD characterization showed that the formation of the h-MoO_3_ polymorph occurred even without triggering the mixing of the precipitating agent with AHM solution inside the droplets by ultrasonication, indicating that HNO_3_ diffused spontaneously into ME droplets. However, since smaller rods (average length of about 450 nm vs. 650 nm) with similar aspect ratio (average AR of 0.3) were obtained without applying US after acid addition, as compared to rods synthesized with the same synthetic conditions and carrying out the US step (Appendix A), it could be argued that, without the US trigger, a slower diffusion of HNO_3_ occurred. Thus, US were not mandatory for the penetration of the added dispersed phase (HNO_3_) into the droplets, which occurs anyway, but their application likely acted as a prompt for a more effective mixing of the precursor and the precipitating agent in the confined space of ME droplets, facilitating and accelerating it. Moreover, it was observed that also the aggregation of MoO_3_ rods was affected by the different mixing approach of Mo precursor and the precipitating agent, and this effect could likely be ascribed to different kinetics of particles formation. In particular, while rods obtained by applying an US step after the acid addition did not display any regular aggregation, the formation of flower-like structures constituted by the aggregation of long micro-rods with hexagonal section developing from the center was observed for samples obtained without US after acid addition (Figure 5a,b). Literature reports of hierarchical flower-like architectures of h-MoO_3_ rods were found, despite being usually obtained under heat treatment (70–120 °C) [16,17,19,31,40]. Chithambararaj et al. proposed that their formation occurred through an oriented attachment mechanism of the 1D hexagonal structures: a spontaneous self-assembly of hexagonal rods occurred by strong interparticle forces, leading to an oriented anisotropic growth of branched flower-like structures [16]. However, it was noticed that, by running the reaction in ME without the US step after acid addition for a longer period of time (48 h vs. 24 h), flower-like aggregates were not observed anymore, and disordered, not aggregated and elongated rod-like structures were obtained (Figure 5c). This evidence was found in agreement with the hypothesized slower reaction kinetics in ME droplets without the external trigger of US, that provided effective and rapid mixing of the reactants.

Finally, in order to exclude the role of US on the morphology and aggregation of MoO_3_ rods obtained in ME, besides enhancing and accelerating the diffusion of the acid into ME droplets, ultrasounds were applied also to a batch synthesis, after the acid addition, with the same experimental conditions employed for ME synthesis. TEM investigations of the batch samples obtained by applying or not applying US after acid addition showed that US-treated batch h-MoO_3_ rods were smaller than non-treated h-MoO_3_ (average rod length of 1250 nm vs. 1700 nm, same AR of about 0.4, see Appendix A), but always significatively larger than ME rods. Thus, it could be concluded that the ME particles size was chiefly controlled by the spatial confinement provided by the ME droplets, while the US effect was negligible.

#### 3.1.4. Effect of Reaction Time

To elucidate the formation mechanism of h-MoO_3_ rods by ME and batch approaches, with and without applying US after acid addition, the ex situ XRD patterns of intermediates at different reaction times (i.e., 5 min, 1, 3, 6, and 18 h) were compared (Figure 6). With all synthetic approaches, reaction times shorter than 18 h did not lead to the crystallization of any MoO_3_ polymorphs. The formation of several ammonium polyoxomolybdates with different nuclearities, such as (NH_4_)_4_(Mo_8_O_26_)(H_2_O)_4_, (NH_4_)_6_(Mo_9_O_30_)(H_2_O)_5_, (NH_4_)_2_(Mo_3_O_10_) and (NH_4_)_2_(Mo_4_O_13_), characterized by quite big unit cells (i.e., most intense reflections at low Bragg angles, 2*θ* < 15°) and constituted by MoO_6_ octahedra sharing vertices or edges [50,51], was instead evidenced. Such a finding was expected, as it is well-known from the complex chemistry of polyoxometalates that the acidification of a [MoO_4_]^2−^ solution gives rise to polyoxomolybdates, which increase in nuclearity as the pH of the solution decreases, until the most stable product (i.e., the metal oxide itself) is formed [38]. By batch approach, crystalline h-MoO_3_ was obtained after 18 h. On the other hand, 18 h ME samples diffractograms displayed an amorphous pattern (25–35° 2*θ*), and superimposed less intense principal reflections of h-MoO_3_ only when US were applied (Figure 6a). This observation further supported the hypothesis of a slower (i.e., lower, after the same reaction time) diffusion of HNO_3_ into the AHM containing-ME droplets without US application. After 24 h, the crystallinity of the samples was further improved, and a decrease in the amorphous percentage was observed for ME samples. These observations (i.e., the XRD patterns evolution as a function of reaction time and the final formation of a metastable polymorph instead of the thermodynamically stable one) is consistent with a kinetic polymorph control and with the Ostwald’s rule of stages, predicting the phase that is energetically closest to the initial state to be kinetically most readily accessible [52]. Indeed, in the case of kinetic polymorph control, crystallization does not follow a single-step pathway, but rather a sequential process involving structural and compositional modifications of amorphous and/or crystalline intermediates. According to the Ostwald’s “step rule”, the least dense phase is formed first, and gradually transforms into the next dense phase until the densest one (i.e., usually also the most stable phase) is formed. This kinetic transformation cascade occurs in order of increasing thermodynamic stability [53]. Considering the Ostwald step rule, and considering that both polyoxomolybdates and MoO_3_ are constituted by the same primary unit of MoO_6_ octahedra differently arranged, it is likely that the investigated crystallization is a stepwise process occurring through subsequent transformations of kinetically accessible transient phases into the final polymorph. These transformations involve the condensation and rearrangement of MoO_6_ octahedra from the polyoxomolybdates phases into the most stable phase (h-MoO_3_).

The evolution of the size and morphology of ME and batch MoO_3_ rods as a function of time (18 h vs. 24 h) was investigated by TEM microscopy (Appendix A). As expected, and supported from XRD patterns evolution, an increase of the average rod length by extending the reaction time (ME: from about 450 nm to 650 nm; batch: from about 1050 nm to 1700 nm) was observed. However, in the case of ME samples, the average AR (from 0.2 to 0.3) and the center values of the size distribution increased, indicating that the rods mostly grew more in width than in length from 18 to 24 h.

### 3.2. Time-Resolved in Situ SAXS/WAXS Study of MoO_3_ Crystallization by Inverse Miniemulsion and Batch Approaches

To shed light on the time evolution of the crystallization of h-MoO_3_, and further deepen the acquired knowledge on the effect of the synthesis parameters and approaches (i.e., batch vs. confined space ME, and application vs. non-application of US) investigated and discussed in Section 3.1, a time-resolved in situ simultaneous SAXS/WAXS study was performed at the SAXS beamline at Elettra Sincrotrone Trieste. A continuous flow-setup (Figure 1) was built at the beamline, efficiently reproducing the laboratory procedures, and at the same time providing the flexibility for designing different series of experiments by exploiting both synthetic approaches, and enabling the investigation of the entire process without blind periods of time, thanks to the possibility of remotely controlling each reaction step. Three series of experiments were performed, exploiting: (i) the ME approach (first series of experiments); (ii) the ME approach without applying US after acid addition (second series of experiments); and (iii) the batch approach (third series of experiments). Since the molar ratio between the Mo precursor and the precipitating agent (HNO_3_) was observed to play the most relevant role on determining MoO_3_ crystallinity, size and morphology (Section 3.1), within each series of experiments, the AHM:HNO_3_ molar ratio was varied between 1:10 and 1:25 mol and the AHM concentration in aqueous solution was kept constant at 0.20 M. The SAXS and WAXS profiles of the reaction mixture were followed simultaneously as a function of time up to 1–3 h (depending on the observed variation of profile features), setting as t = 0 s the start of acid injection.

#### 3.2.1. First Series: Miniemulsion Syntheses

As expected, during the first steps of the miniemulsion synthesis, i.e., (i) formation of AHM containing-ME by US (2 min), and (ii) conc. nitric acid addition (set as t = 0 s), an increase in the SAXS signal intensity was observed, due to the increased number of scattering objects. Subsequently, during the application of a further step of US (step iii, 2 min), the intensity of the scattering profile decreased, proving the penetration of HNO_3_ into ME droplets (i.e., reduced total number of scattering objects). Finally, a further increase in scattering intensity was observed due to the occurrence of dehydration and polycondensation reactions of molybdates in acidic medium (i.e., formation of Mo-based particles with higher electron density with respect to surfactant-stabilized water droplets dispersed in cyclohexane), concurrent to the formation of reflections in the higher q SAXS region (2–7 nm^−1^, insets in Figure 7a,b). The different widths of the Bragg reflections, their relative position and relative rate of growth suggested that the observed diffractograms consisted of the superimposition of different individual patterns ascribable to different species and/or derived from different origins (e.g., from a long-range order in a crystalline structure or from the ordered array of primary particles). Indeed, at least two patterns having hexagonal symmetry (i.e., whose relative position followed the relation 1, $\sqrt{3}$, 2, $\sqrt{7}$, 3 [54], more evident with higher amounts of HNO_3_) were recognized, characterized by broad peaks positioned at *q_0,1_* = 2.33 nm^−1^, *q_1,1_* = 4.04 nm^−1^, *q_2,1_* = 4.66 nm^−1^, *q_3,1_* = 6.16 nm^−1^, *q_4,1_* = 6.99 nm^−1^ and *q_0,2_* = 3.33 nm^−1^, *q_1,2_* = 5.72 nm^−1^, *q_2,2_* = 6.66 nm^−1^. Being ruled out as the possible role of the surfactant in the origin of these patterns (see Section 3.2.3), they could be ascribed to the formation of two hexagonal mesophases (e.g., due to rods arranged in a hexagonal array), through a nonclassical crystallization pathway [55]. Additionally, sharper reflections, not displaying any specific spatial relationship, were observed at 2.89 nm^−1^, 3.69 nm^−1^, 3.84 nm^−1^, 4.01 nm^−1^, 4.96 nm^−1^, 6.05 nm^−1^, 6.29 nm^−1^, and 6.56 nm^−1^.

Interestingly, by increasing the relative amount of acid in the synthesis in ME (Figure 7a,b), the SAXS sharp peaks disappeared accordingly, showing only two weak peaks at 3.69 nm^−1^ and 4.96 nm^−1^ (the more the acid, the less intense the peaks, almost undetectable at AHM:HNO_3_ 1:25 mol). Moreover, the broad peaks (i.e., the two hexagonal patterns starting with *q_0,1_* = 2.33 nm^−1^ and *q_0,2_* = 3.33 nm^−1^) grew faster by increasing the amount of acid, both in absolute intensity and relatively to the sharper peaks (the more the acid, the more intense the peaks). Indeed, after 250 s (at the end of step iii, i.e., US), while at AHM:HNO_3_ 1:10 mol the sharper peaks stood out in the pattern, at AHM:HNO_3_ 1:25 mol they were barely detectable compared to the broader ones (Figure 7a). Moreover, by following the evolution of the pattern for 5000–10,000 s (Figure 7b) it could be appreciated that, with higher amounts of acid, the observed peaks firstly grew, but subsequently their intensity decreased, suggesting the formation and disappearance of a reaction intermediate(s), more stable (i.e., stable for a longer period of time) in the experiments performed with lower amounts of acid (1:10 and 1:15 mol). Such intermediate species could likely be identified in a (mixture of) polyoxomolydate(s), known intermediates in the polycondensation reactions occurring in acidified Mo(VI) solution [50].

The simultaneously-acquired WAXS measurements (16–22 nm^−1^, i.e., 23–31° 2*θ*), shown in Figure 7c, enabled us to shed light on the nature of the intermediate species, and to support the ex situ time-resolved investigations of the formation of h-MoO_3_ discussed above (Section 3.1.4): the crystallization of the metastable h-MoO_3_ polymorph was achieved after several hours of reaction at room temperature, and the species present in suspension in the first 1–3 h of reaction were intermediates. Indeed, the characteristic reflections of h-MoO_3_ (in the acquired *q*-range: 18.1, 20.5 and 21.6 nm^−1^, ascribed to the (211), (300) and (204) reflections, respectively, see Appendix A in the SI), not observed in the time-resolved in situ experiments within the 1–3 h of acquisition, were displayed in the WAXS pattern of a 24 h reaction mixture. Moreover, the amount of HNO_3_ in the reaction mixture was found to strongly affect the crystallization pathway of MoO_3_, obtaining the transitory formation of crystalline species with higher AHM:HNO_3_ molar ratio (1:10 mol, about 5000 s, i.e., 1 h-1 h and 30 min), as shown in Figure 7c. Given the plethora of sharp reflections observed in a small q range, it was likely that the acquired patterns resulted from the crystallization of a mixture of compounds, in agreement with the hypothesized formation of several polyoxomolybdates species. By following the evolution of the profiles for a longer period of time (until about 9000 s, i.e., 2 h and 30 min), the intensity of these reflections decreased until only fewer and larger reflections could be appreciated. From this evidence, it could be supposed that the sharp reflections observed in the WAXS range and the sharp reflections formed at lower angle (SAXS region) are ascribable to the same intermediate species, that crystallized with a relatively large unit cell. Their absence in the WAXS patterns acquired with higher amounts of nitric acid (AHM:HNO_3_ 1:20 and 1:25 mol) was found consistent with the SAXS profile evolutions discussed above. Moreover, the absence of MoO_3_ polymorphs reflections as well supported the hypothesis that, in these conditions, the formation of MoO_3_ occurred likely through an amorphous intermediate, obtaining the final hexagonal polymorph after 18–24 h of reaction, through arrangement and densification towards the final structure, according to the Ostwald’s rule of stages [52]. Finally, as no broad reflections were found in the WAXS region, the broad SAXS reflections could likely be ascribed to an ordered hexagonal array of primary cylindrical particles, thus supporting the hypothesis of a non-classical mechanisms of crystallization, involving the oriented aggregation of primary particles into the formation of a mesoscale assembly [55,56].

#### 3.2.2. Second Series: Miniemulsion Syntheses without US Step after Acid Addition

As expected, the time-resolved evolution of the SAXS profiles of the second series of experiments (Figure 8a,b) was comparable to the one of the first series (i.e., differing in the application of US after acid addition (Figure 7a,b)), i.e., overall scattering intensity increase and reflections growth in the high q region. However, slower features variations were observed, confirming the previously discussed hypotheses based on ex situ evidence (i.e., reactants mixing occurred only via acid diffusion into ME droplets, without an external trigger (i.e., US) that ensured a more rapid and efficient interchange of materials).

Analogously to the experiment in ME with the US step and the lower amount of acid (1:10 mol, Figure 7a,b), the early growth of the sharp peaks in the SAXS region was observed in all the experiments of the second series (AHM:HNO_3_ from 1:10 to 1:25 mol, Figure 8a,b). This could likely be ascribed again to the mixing of reactants occurring only by acid diffusion into the droplets: being it slow, during the first acquisitions (250 s) only a small portion of acid should have diffused into the droplets (i.e., the actual relative amount of acid in the droplets was lower than the nominal one). After a longer period of observation, the same conclusions drawn for the first series of experiments could be set: the higher the amount of acid, the higher the relative intensity of the broader peaks with respect to the sharper ones. Interestingly, the disappearance of the initially formed sharp peaks was observed within 1.5 h for the experiments with AHM:HNO_3_ of 1:20 and 1:25 mol, while being their intensity clearly lowered for the 1:15 mol one.

The time-resolved evolution of the simultaneously-acquired WAXS profiles (Figure 8c) supported these observations. Indeed, the crystallization of transitory crystalline species detected in the first series of experiments (Figure 7c) was clearly evidenced with low amounts of acid (AHM:HNO_3_ of 1:10 and 1:15 mol), and, although less significatively, also by increasing the amount of nitric acid over 15 times the moles of AHM, in contrast to the first series of experiments, where no reflection growth was appreciated with AHM:HNO_3_ molar ratios of 1:20 and 1:25. This finding further supported the effect of US of dramatic enhancement of the acid penetration into ME droplets, suddenly increasing the concentration of $H^{+}$ in the reaction environment (ME droplets) and shifting the equilibrium of the reaction towards the products. On the other hand, when diffusion of HNO_3_ through the continuous phase into ME droplets (second series of experiments) governed alone the mixing of AHM with the precipitating agent, the polycondensation reactions occurred more slowly and intense diffraction of polyoxomolybdates species could be appreciated also for 1:15 mol experiment, and less markedly for 1:20 and (in the first frames of) 1:25 mol ones. Finally, the stability of these intermediates for a longer period of time with respect to the first series of experiments was observed.

#### 3.2.3. Third Series: Batch Syntheses

Analogously to the two ME series of experiments, the time-resolved evolution of SAXS profile of the batch reaction mixture displayed an increase in the total scattering intensity when nitric acid was added to AHM aqueous solution and the appearance of reflections in the high *q* range of the SAXS region. However, the obtained pattern resulted different in number, position, and widths of the Bragg peaks with respect to ME one. In particular, the broad reflections (the two hexagonal patterns starting with *q_0,1_* = 2.33 nm^−1^ and q_0,2_ = 3.33 nm^−1^) were not found, and the lowest q peak was observed at 2.89 nm^−1^. On the contrary, the sharper reflections obtained by ME approaches (at 2.89 nm^−1^, 3.69 nm^−1^, 3.84 nm^−1^, and so on, listed above) were observed, indicating the formation of one or more common species. In addition to these, the batch approach pattern displayed other sharp reflections at *q* = 4.19 nm^−1^, 4.40 nm^−1^, 4.50 nm^−1^, 4.90 nm^−1^, 5.08 nm^−1^, 5.70 nm^−1^, 5.90 nm^−1^, 6.76 nm^−1^, and 6.81 nm^−1^, not showed in the ME ones.

By following the evolution of the SAXS profile feature as a function of reaction time and AHM:HNO_3_ molar ratio (Figure 9a,b), it was observed that the sharp reflection found only in the batch series (*q* = 4.19 nm^−1^, 4.40 nm^−1^, 4.50 nm^−1^, and so on, listed above) were likely ascribable to a more reactive reaction intermediate(s), as they first grew and then disappeared, while the other peaks did not (or were observed for a longer period of time, in the case of high amounts of acid, *see below*). A variation of the relative amount of HNO_3_ did not affect the relative growth of the peaks during the first acquisitions (Figure 9a, t = 0–600 s), found for the ME series instead (Figure 7a). However, considering longer reaction times (until 5500–6000 s, Figure 9b), it was evident that, with acid amounts above 1:20 mol, the intensity of all the Bragg peaks of the pattern decreased until no peaks were found at all (1:25 mol, 7000 s). Since a similar trend was outlined for the ME patterns obtained at AHM:HNO_3_ 1:20 or 1:25 mol (first and second series, Figure 7b and Figure 8b), i.e., the sharper peaks observed also in batch conditions almost disappeared after 5000–7000 s, it could be assumed that, in both conditions, the same intermediate, more stable in less acidic conditions (i.e., AHM:HNO_3_ 1:10 and 1:15 mol), was formed. In addition to this species formed under both conditions, by both approaches at least another product (whose nature is dependent on the synthetic approach) was formed, and it was likely (i) ordered microdomains in hexagonal arrays in ME (characterized by two superimposed patterns of broad reflections with hexagonal symmetry starting with *q_0,1_* = 2.33 nm^−1^ and *q_0,2_* = 3.33 nm^−1^) and (ii) a more reactive intermediate species leading to sharp reflections in the batch one (the ones at *q* = 4.19 nm^−1^, 4.40 nm^−1^, etc., listed above, that readily formed and disappeared).

As for the two ME series of experiments, the conclusions derived so far were confirmed by the simultaneously-acquired WAXS profiles (Figure 9c). In particular, the crystallization of MoO_3_ is a stepwise process that occurs through the formation of likely a mixture of polyoxomolybdates as reaction intermediates, more stable in less acidic conditions and that gradually go through a phase-transformation from a crystalline polymorph to an amorphous (no reflections in the WAXS region) material and ultimately to h-MoO_3_ (after 18–24 h, see above). As mentioned above, these observed stepwise transformations of the product are consistent with the Ostwald’s rule of stages, and likely occur through the gradual condensation and rearrangement of MoO_6_ octahedra, primary units of all the observed transient species and the final product h-MoO_3_. Interestingly, the synthetic approach was found to affect the crystallization pathway, obtaining a more controlled mixing of the Mo (VI) precursor and the precipitating agent in ME droplets (first and second series), and in particular exploiting the spontaneous diffusion of HNO_3_ into ME droplets (second series), with respect to the macroreactor approach (batch approach, third series). Moreover, ME droplets seemed to promote a non-classical crystallization pathway involving the oriented aggregation of primary particles into a hexagonal array (i.e., likely a mesocrystal), while preventing the formation of a more reactive reaction intermediate that was observed in batch conditions. From this evidence, it could also be argued that the spatial confinement plays a role in altering the intermediates stability and their transformation to the next stable species according to Ostwald’s step rule, known to depend on the free energies of activation of nucleation in different environments [53].

As mentioned above, the observations related to the origin of the broad peaks interrelated by a hexagonal symmetry found in the ME series of experiments (first and second series) and not in the batch one (third series) were further confirmed by additional ME experiments performed using a different surfactant to stabilize ME droplets (i.e., PGPR instead of Span80), and AHM:HNO_3_ molar ratios of 1:10 and 1:25. The aim of these experiments was to rule out the possibility of ascribing the observed ME pattern to the particular surfactant employed (Span80), forming ordered aggregates in solution (e.g., micelles or liquid crystals), even if in concentration below the critical micelle concentration. The same pattern was found with PGPR as surfactant and both reactants’ molar ratios (see Appendix A): the differences in the ME and batch patterns were completely ascribed to the different synthetic approaches, affecting the crystallization pathway of MoO_3_. Indeed, it is worth reiterating that confining the synthesis of MoO_3_ within ME droplets likely enables an earlier-oriented aggregation of primary amorphous units into hexagonal arrays, that further condensate into the target hexagonal polymorph of MoO_3_.

## 4. Conclusions

The room temperature synthesis of h-MoO_3_ in the confined space of miniemulsion nanodroplets and in a macro-reactor (batch approach) was comprehensively investigated, exploiting the dehydration and polycondensation reactions occurring in acidified solutions of Mo(VI). A systematic screening of the reaction parameters (i.e., AHM concentration, AHM:HNO_3_ molar ratio, application of ultrasounds, and reaction time) was carried out, and their effect on the synthetic pathway and final outcome, as well as the effect of ME spatial confinement, was thoroughly evaluated. Polycrystalline MoO_3_ rods with hexagonal cross-sections and different size (nano- to micro-rods), aspect ratio and aggregation as a function of the synthetic approach and experimental parameters were obtained. Notably, a relevant role of ME spatial confinement was observed on the size and shape of h-MoO_3_, obtaining smaller and more elongated rods by ME approach, as compared to batch MoO_3_. Moreover, by both ex situ and in situ time-resolved investigations, the crystallization of MoO_3_ by ME and batch approaches was demonstrated to occur as a stepwise process, following the Ostwald’s rule of stages, encompassing (i) the initial formation of ammonium polyoxomolybdates, (ii) a subsequent phase-transformation to an amorphous material, and (iii) the final condensation to h-MoO_3_. Interestingly, it was observed that confining the synthesis in ME droplets promoted a nonclassical crystallization pathway involving the oriented aggregation of primary particles into hexagonal arrays (i.e., likely mesocrystals), while preventing the formation of a more reactive reaction intermediate that was observed in batch conditions. It was thus argued that the space confinement plays a role in altering the stability of intermediate products and their interconversion, according to the Ostwald’s rule of stages.

## Figures and Tables

**Figure 1 nanomaterials-13-01046-f001:**
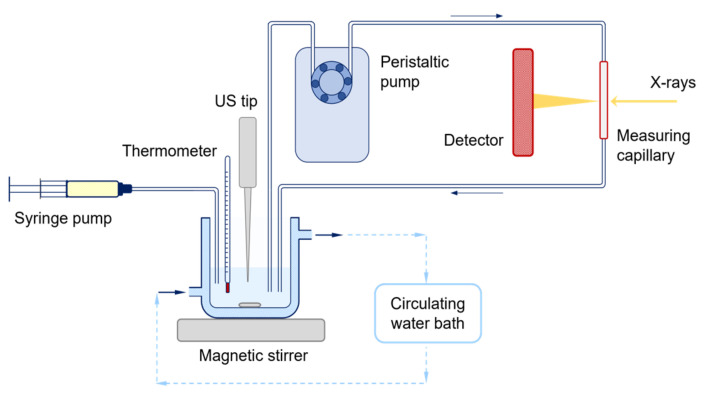
Schematic representation of the continuous-flow setup built at the SAXS beamline at Elettra Sincrotrone Trieste for the in situ time-resolved SAXS/WAXS study of MoO_3_ synthesis.

**Figure 2 nanomaterials-13-01046-f002:**
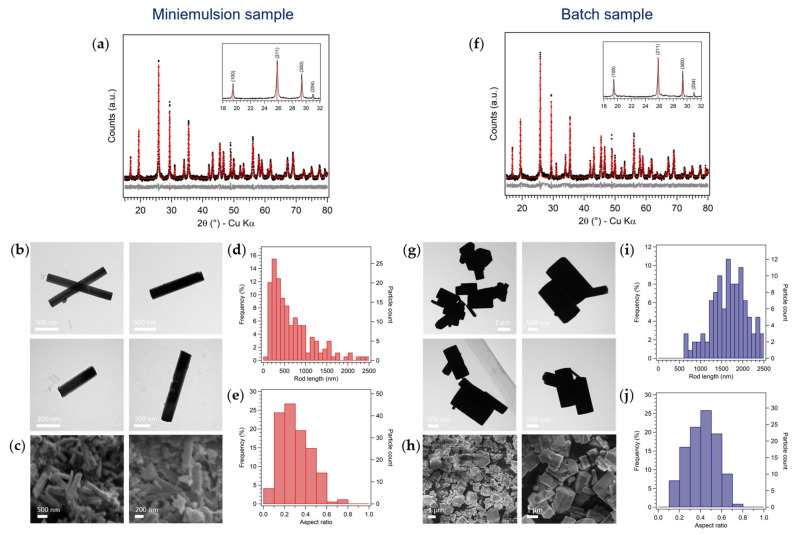
Characterization of (**a**–**e**) ME and (**f**–**j**) batch samples ([AHM] = 0.20 M, AHM:HNO_3_ 1:10 mol, 24 h): (**a**,**f**) XRD fittings, insets: XRD patterns (18–32° 2θ) superimposed with reference database pattern (red sticks); (**b**,**g**) TEM and (**c**,**h**) SEM micrographs; histograms representing the (**d**,**i**) rod length and (**e**,**j**) aspect ratio distributions of samples, estimated from TEM micrographs.

**Figure 3 nanomaterials-13-01046-f003:**
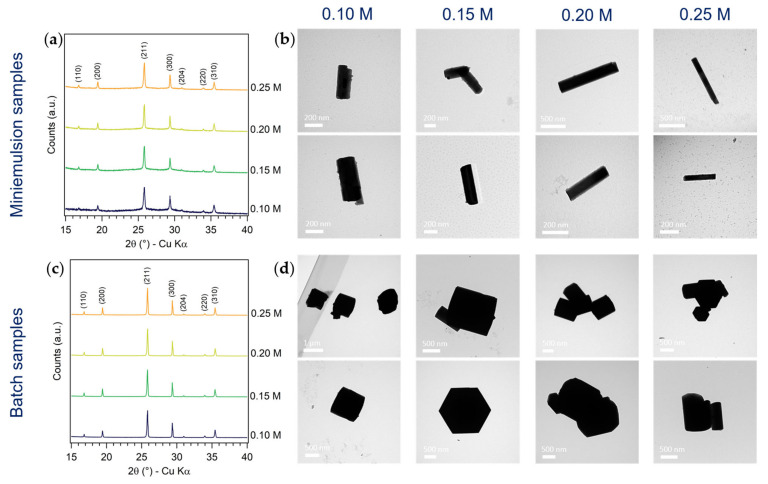
Characterization of (**a**,**b**) ME and (**c**,**d**) batch samples synthesized at different AHM concentrations (0.10, 0.15, 0.20 and 0.25 M) and constant AHM:HNO_3_ molar ratio (1:10 mol) and reaction time (24 h): (**a**,**c**) comparison of XRD patterns (15–60° 2θ), (**b**,**d**) TEM micrographs.

**Figure 4 nanomaterials-13-01046-f004:**
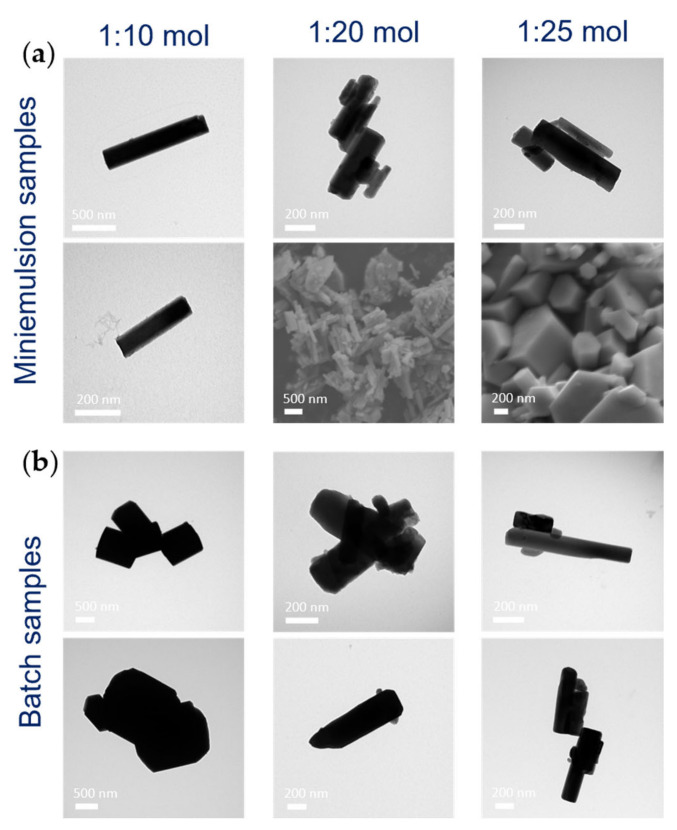
TEM and SEM micrographs of (**a**) ME and (**b**) batch samples synthesized with different AHM:HNO_3_ molar ratios (1:10, 1:20, and 1:25 mol) and constant AHM concentrations ([AHM] = 0.20 M) and reaction time (24 h).

**Figure 5 nanomaterials-13-01046-f005:**
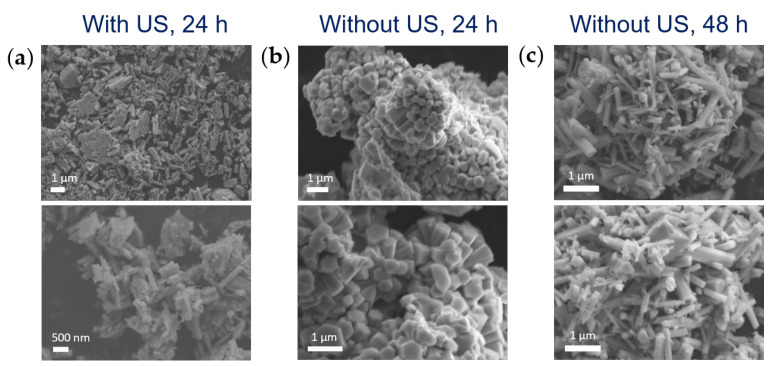
SEM images of ME samples synthesized by (**a**) applying or (**b**,**c**) not applying US after acid addition and running the reaction for (**a**,**b**) 24 h or (**c**) 48 h. [AHM] = 0.20 M; AHM: HNO_3_ 1:20 mol.

**Figure 6 nanomaterials-13-01046-f006:**
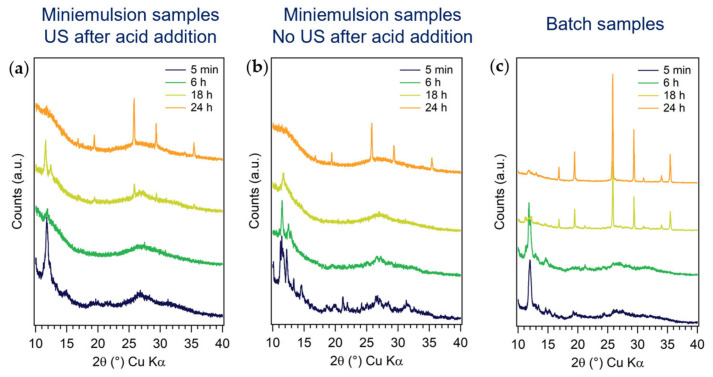
Comparison between XRD patterns of samples obtained in ME (**a**) with and (**b**) without applying US after acid addition and (**c**) in batch, after different reaction times (5 min, 6 h, 18 h, and 24 h). [AHM] = 0.20 M; AHM: HNO_3_ 1:10 mol.

**Figure 7 nanomaterials-13-01046-f007:**
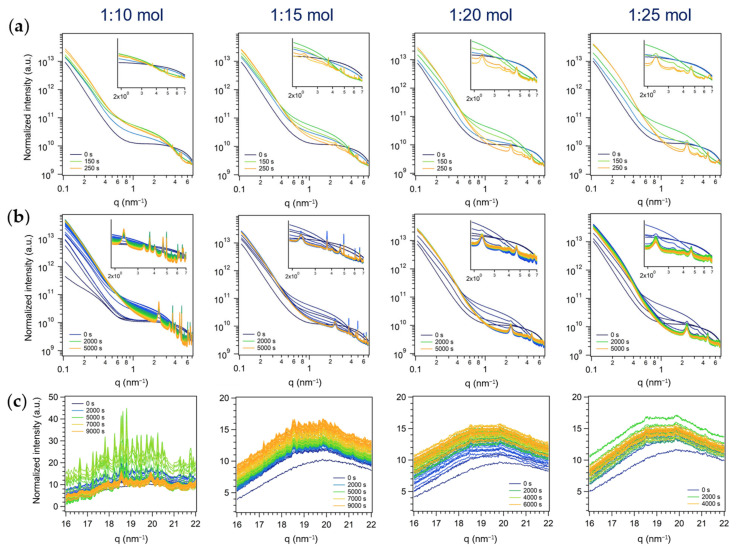
In situ time-resolved (**a**,**b**) log-log plots of SAXS and (**c**) simultaneous WAXS patterns of the series of experiments performed in ME with different AHM:HNO_3_ molar ratios (first series of experiments). A SAXS pattern every 50 s from the start of acid addition (t = 0 s) to (**a**) the end of US (step iii) (t = 250 s) and to (**b**) 5000 s. Insets: zoom on Bragg peaks (1.9–7 nm^−1^). (**c**) A WAXS pattern every 100 s from the start of acid addition (t = 0 s) to the end of acquisition.

**Figure 8 nanomaterials-13-01046-f008:**
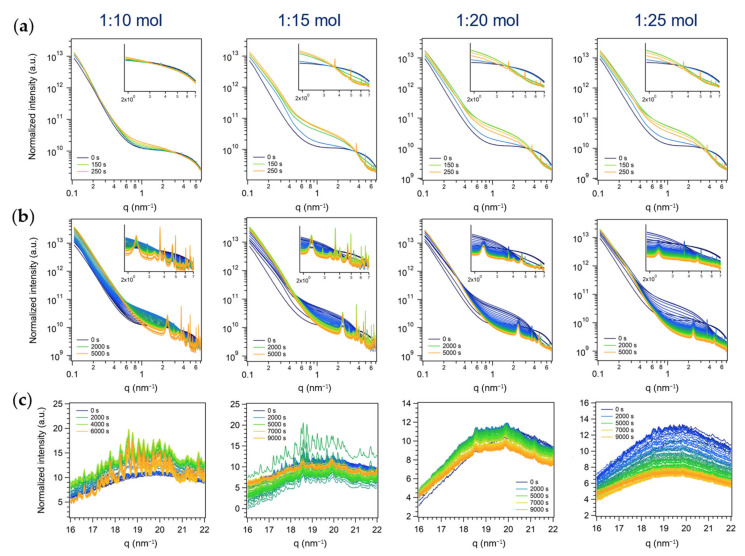
In situ time-resolved (**a**,**b**) log-log plots of SAXS and (**c**) simultaneous WAXS patterns of the series of experiments performed in ME without the US step with different AHM:HNO_3_ molar ratios (second series of experiments). A SAXS pattern every 50 s from the start of acid addition (t = 0 s) to (**a**) 250 s (i.e., corresponding to the end of US step for the first series of experiments) and to (**b**) 5000 s. Insets: zoom on Bragg peaks (1.9–7 nm^−1^). (**c**) A WAXS pattern every 100 s from the start of acid addition (t = 0 s) to the end of acquisition.

**Figure 9 nanomaterials-13-01046-f009:**
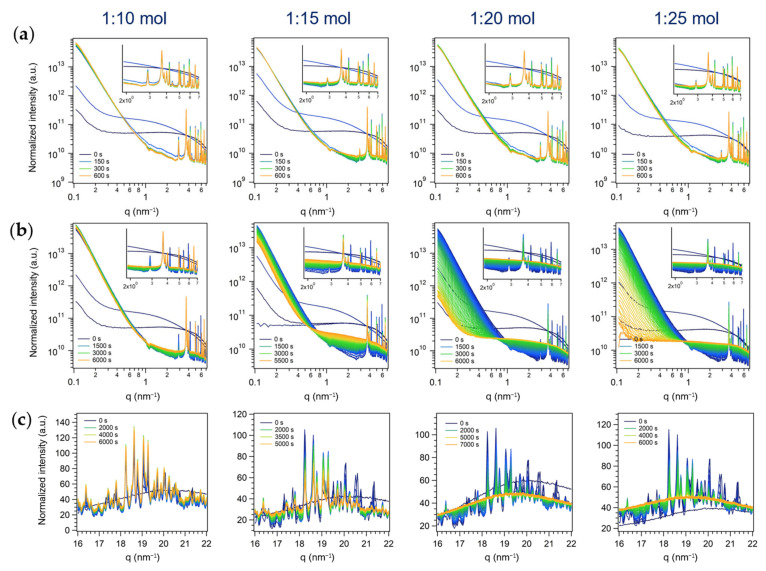
In situ time-resolved (**a**,**b**) log-log plots of SAXS and (**c**) simultaneous WAXS patterns of the series of experiments performed in batch with different AHM:HNO_3_ molar ratios (third series of experiments). A SAXS pattern (**a**) every 50 s from the start of acid addition (t = 0 s) to 600 s and (**b**) every 100 s from the start of acid addition (t = 0 s) to 5500–6000 s. Insets: zoom on Bragg peaks (1.9–7 nm^−1^). (**c**) A WAXS pattern every 100 s from the start of acid addition (t = 0 s) to the end of acquisition.

## Data Availability

Additional information is available from the authors.

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
