# Peer review of "Exploring the Role of Miniemulsion Nanodroplet Confinement on the Crystallization of MoO3: Morphology Control and Insight on Crystal Formation by In Situ Time-Resolved SAXS/WAXS"

_nanomaterials, 2023, doi:10.3390/nano13061046_

Round 1

Reviewer 1 Report

The authors present a concept of academic interest. The article is concise and to the point.

The authors should include counts/intensity of XRD in Figure 3 and Figure 6

Reviewer 2 Report

Interesting discoveries regarding the impact of the preparation techniques on the morphology of MoO3 crystals are detailed in the publication by Gross et al.
The combination of characterization techniques (XRD, TEM, IR, and DLS) used by the authors provides great support for the discussion of the experimental results.

Several minor comments that should be addressed by authors in their revised version:

1. Define BL where first introduced (p. 3, line 122).

2. p.4 Experimental section: ATR spectra were acquired in the 4000-400 cm-1 range. But on p. 6 frequencies in the 4500-500 cm-1 range were discussed and shown in Fig. S1. Also, the characteristic frequencies which are discussed in the main text should be labelled on the IR spectra provided in Fig. S1.

3. There is no agreement between the discussion on p. 6, lines 247 - 251 and the figures indicated (Fig. 2a,c and 2b,d should be 3a,c and 3b,d).

4. The authors stated that they replaced Span 80 with PGPR to examine the influence of the surfactant as well, but no experimental support was offered.

Reviewer 3 Report

In this manuscript, authors used two different approaches (miniemulsion confinement growth and traditional wet chemistry process) to prepare MoO3 material by controlling reaction molar ratio, precursor concentration and reaction time. XRD, IR, S/TEM, DLS were be used to characterize the structure, composite and size distribution. And in situ time-resolved SAXS/WAXS was used to monitor the crystallization pathway of MoO3 in both synthetic conditions. This work is of interest and significant, I would like to recommend this manuscript for publication in Naonomaterials after minor revision.

1. In introduction section, what is the advantage between miniemulsion and traditional chemical approach for the preparation of MoO3?

2. How to compare present results to previous work reported by other researchers via microemulsion method?

3. Note that batch samples showed large particle size, how to compare this case with previous work?

4. How to explain slight change on aspect ratio?

5. Influence of reaction time is very important for the preparation of MoO3, please emphasize the formation mechanism of final product.

6. It is clear to see concentration effect of AHM on particle shape and size, please try to explain these causes.

7. Please simplify and shorten conclusions.

Reviewer 4 Report

Referee Report

On the paper “ Exploring the role of miniemulsion nanodroplet confinement on the crystallization of MoO3: morphology control and insight on crystal formation by in situ time-resolved SAXS/WAXS “ (nanomaterials-2264354) by the authors Francesca Tajoli, Maria Vittoria Massagrande, Rafael Muñoz-Espí and Silvia Gross submitted to the Nanomaterials

This is interesting paper. It reports the synthesis of MoO3 nano- and microrods with hexagonal section in inverse miniemulsion droplets and batch conditions, evaluating the effects of spatial confinement offered by miniemulsion droplets on their crystallization. Several synthetic parameters were systematically screened and their effect on the crystal structure of h-MoO3, as well as on its size, size distribution and morphology, were investigated. The obtained experimental results are reliable without any doubts. However, I have some questions and additions. I would like to note a few points to improve the paper before it can be published:

1.    The authors should give examples in 1. Introduction of the formation of nanoobjects by other different technique:

(1). A.V. Trukhanov, S.S. Grabchikov, S.A. Sharko, S.V. Trukhanov, E.L. Trukhanova, O.S. Volkova, A. Shakin, Magnetotransport properties and calculation of the stability of GMR coefficients in CoNi/Cu multilayer quasi-one-dimensional structures, Mater. Res. Express 3 (2016) 065010. https://doi.org/10.1088/2053-1591/3/6/065010.

(2). S.A. Sharko, A.I. Serokurova, T.I. Zubar, S.V. Trukhanov, D.I. Tishkevich, A.A. Samokhvalov, A.L. Kozlovskiy, M.V. Zdorovets, L.V. Panina, V.M. Fedosyuk, A.V. Trukhanov, Multilayer spin-valve CoFeP/Cu nanowires with giant magnetoresistance, J. Alloys Compd. 846 (2020) 156474. https://doi.org/10.1016/j.jallcom.2020.156474.

2.    The authors should mention in 1. Introduction some results of the preparation and investigation of other oxides promising for practical applications:

(3). S.V. Trukhanov, A.V. Trukhanov, A.N. Vasiliev, H. Szymczak, Frustrated exchange interactions formation at low temperatures and high hydrostatic pressures in La0.70Sr0.30MnO2.85, J. Exp. Theor. Phys. 111 (2010) 209-214. https://doi.org/10.1134/S106377611008008X.

(4). I.Z. Zhumatayeva, I.E. Kenzhina, A.L. Kozlovskiy, M.V. Zdorovets, The study of the prospects for the use of Li0.15Sr0.85TiO3 ceramics, J. Mater. Sci.: Mater. Electron. 31 (2020) 6764-6772. https://doi.org/10.1007/s10854-020-03234-9.

3.    The authors should give information in 1. Introduction on the importance of the oxide based nanocomposites promising for practical applications:

(5). A. Kozlovskiy, K. Egizbek, M.V. Zdorovets, M. Ibragimova, A. Shumskaya, A.A. Rogachev, Z.V. Ignatovich, K. Kadyrzhanov, Evaluation of the efficiency of detection and capture of manganese in aqueous solutions of FeCeOx nanocomposites doped with Nb2O5, Sensors 20 (2020) 4851. https://doi.org/10.3390/s20174851.

(6). M.A. Almessiere, Y. Slimani, N.A. Algarou, M.G. Vakhitov, D.S. Klygach, A. Baykal, T.I. Zubar, S.V. Trukhanov, A.V. Trukhanov, H. Attia, M. Sertkol, I.A. Auwal, Tuning the structure, magnetic and high frequency properties of Sc-doped Sr0.5Ba0.5ScxFe12-xO19/NiFe2O4 hard/soft nanocomposites, Adv. Electr. Mater. 8 (2022) 2101124. https://doi.org/10.1002/aelm.202101124.

4.    The proposed 6 papers should be inserted in References.

The paper should be sent to me for the second analysis after the moderate revisions.

Round 2

Reviewer 4 Report

Referee Report

On the paper “ Exploring the role of miniemulsion nanodroplet confinement on the crystallization of MoO3: morphology control and insight on crystal formation by in situ time-resolved SAXS/WAXS “ (nanomaterials-2264354-v2) by the authors Francesca Tajoli, Maria Vittoria Massagrande, Rafael Muñoz-Espí and Silvia Gross submitted to the Nanomaterials

Unfortunately, the necessary changes have not been made. I suggest the authors try again to improve their paper. The authors should be more attentive and scrupulous to the suggestions and additions of the reviewer in order to achieve the desired result promptly:

1.    The authors should give examples in 1. Introduction of the formation of nanoobjects by other different technique:

(1). A.V. Trukhanov, S.S. Grabchikov, S.A. Sharko, S.V. Trukhanov, E.L. Trukhanova, O.S. Volkova, A. Shakin, Magnetotransport properties and calculation of the stability of GMR coefficients in CoNi/Cu multilayer quasi-one-dimensional structures, Mater. Res. Express 3 (2016) 065010. https://doi.org/10.1088/2053-1591/3/6/065010.

(2). S.A. Sharko, A.I. Serokurova, T.I. Zubar, S.V. Trukhanov, D.I. Tishkevich, A.A. Samokhvalov, A.L. Kozlovskiy, M.V. Zdorovets, L.V. Panina, V.M. Fedosyuk, A.V. Trukhanov, Multilayer spin-valve CoFeP/Cu nanowires with giant magnetoresistance, J. Alloys Compd. 846 (2020) 156474. https://doi.org/10.1016/j.jallcom.2020.156474.

2.    The authors should mention in 1. Introduction some results of the preparation and investigation of other oxides promising for practical applications:

(3). S.V. Trukhanov, A.V. Trukhanov, A.N. Vasiliev, H. Szymczak, Frustrated exchange interactions formation at low temperatures and high hydrostatic pressures in La0.70Sr0.30MnO2.85, J. Exp. Theor. Phys. 111 (2010) 209-214. https://doi.org/10.1134/S106377611008008X.

(4). I.Z. Zhumatayeva, I.E. Kenzhina, A.L. Kozlovskiy, M.V. Zdorovets, The study of the prospects for the use of Li0.15Sr0.85TiO3 ceramics, J. Mater. Sci.: Mater. Electron. 31 (2020) 6764-6772. https://doi.org/10.1007/s10854-020-03234-9.

3.    The authors should give information in 1. Introduction on the importance of the oxide based nanocomposites promising for practical applications:

(5). A. Kozlovskiy, K. Egizbek, M.V. Zdorovets, M. Ibragimova, A. Shumskaya, A.A. Rogachev, Z.V. Ignatovich, K. Kadyrzhanov, Evaluation of the efficiency of detection and capture of manganese in aqueous solutions of FeCeOx nanocomposites doped with Nb2O5, Sensors 20 (2020) 4851. https://doi.org/10.3390/s20174851.

(6). M.A. Almessiere, Y. Slimani, N.A. Algarou, M.G. Vakhitov, D.S. Klygach, A. Baykal, T.I. Zubar, S.V. Trukhanov, A.V. Trukhanov, H. Attia, M. Sertkol, I.A. Auwal, Tuning the structure, magnetic and high frequency properties of Sc-doped Sr0.5Ba0.5ScxFe12-xO19/NiFe2O4 hard/soft nanocomposites, Adv. Electr. Mater. 8 (2022) 2101124. https://doi.org/10.1002/aelm.202101124.

4.    The proposed 6 papers should be inserted in References.

The paper should be sent to me for the third analysis after the major revisions.
